# Genetic interactions affecting human gene expression identified by variance association mapping

Andrew Anand Brown[1,2], Alfonso Buil[3,4,5], Ana Viñuela[6], Tuuli Lappalainen[3,4,5], Hou-Feng Zheng[7], J Brent Richards[6,7], Kerrin S Small[6], Timothy D Spector[6], Emmanouil T Dermitzakis[3,4,5], Richard Durbin[1]*

[1]Human Genetics, Wellcome Trust Sanger Institute, Cambridge, United Kingdom; [2]NORMENT, KG Jebsen Centre for Psychosis Research, Institute of Clinical Medicine, University of Oslo, Oslo, Norway; [3]Department of Genetic Medicine and Development, University of Geneva, Geneva, Switzerland; [4]Institute of Genetics and Genomics in Geneva, University of Geneva Medical School, Geneva, Switzerland; [5]Swiss Institute of Bioinformatics, Geneva, Switzerland; [6]Department of Twin Research and Genetic Epidemiology, King's College London, London, United Kingdom; [7]Department of Medicine, Human Genetics, Epidemiology and Biostatistics, McGill University, Montreal, Canada

**Abstract** Non-additive interaction between genetic variants, or epistasis, is a possible explanation for the gap between heritability of complex traits and the variation explained by identified genetic loci. Interactions give rise to genotype dependent variance, and therefore the identification of variance quantitative trait loci can be an intermediate step to discover both epistasis and gene by environment effects (GxE). Using RNA-sequence data from lymphoblastoid cell lines (LCLs) from the TwinsUK cohort, we identify a candidate set of 508 variance associated SNPs. Exploiting the twin design we show that GxE plays a role in ~70% of these associations. Further investigation of these loci reveals 57 epistatic interactions that replicated in a smaller dataset, explaining on average 4.3% of phenotypic variance. In 24 cases, more variance is explained by the interaction than their additive contributions. Using molecular phenotypes in this way may provide a route to uncovering genetic interactions underlying more complex traits.

*For correspondence: rd@sanger.ac.uk

## Introduction

The discrepancy between the contribution of known genetic factors to variation of a trait and the estimated total contribution of all genetic variants has become known as 'missing heritability' (*Manolio et al., 2009*). Some of the explanations for this discrepancy are: many common variants with small effects; many rare variants with larger effects; and interactions between genetic variants (epistasis) or between variants and environment (GxE). Here, we focus on the discovery and characterisation of epistasis, by which we mean that the effect of a genetic variant on a trait depends on the genotype at one or more other locations in the genome. Statistically we define this as a joint effect of two loci on a trait, significant beyond the sum of additive effects.

On long time frames, epistasis plays an important role in evolution (*Breen et al., 2012*), and has been used to explain the persistence of deleterious mutations under selection (*Hemani et al., 2013*). Epistasis has frequently been seen in crosses between model organism strains. *Huang et al. (2012)* looked at mapping variants associated with three traits in two distinct *Drosophila* populations and found very little concordance between the results. They postulated that this could be because the

**eLife digest** Every person has two copies of each gene: one is inherited from their mother and the other from their father. These two copies are often not identical because there can be many different variants of the same gene in the human population. Traits (such as height, body mass and risk of disease) vary from one person to the next—and for many traits this variation depends in part on the different gene variants that each person has inherited. Studies seeking to find the differences in DNA that can predict this variation have often assumed that the changes in DNA act on traits independently of the effect of environment and of other genetic variants.

In contrast, studies with animals have shown that some genetic variants can interact to produce a bigger (or smaller) effect than would be expected from simply 'adding together' their individual effects—a phenomenon called epistasis. But how much does epistasis contribute to variation in human traits, if at all? This question has been much disputed, and is difficult to test, not least because of the sheer number of interactions to assess: tens of millions of changes in DNA have been observed in the human genome, and so there are many more than billions of possible combinations of these changes to investigate.

Here, Brown et al. have examined the sequences of all the genes that were expressed in cells taken from a cohort of twins and searched for genetic variants that show these epistatic interactions. By studying gene expression, which can be greatly affected by small changes in the DNA code, Brown et al. were able to identify 508 variants that had a bigger than expected effect on the level of gene expression. This may be a sign that these variants act in combinations: if within one genome a variant increased expression and in another it decreased expression, then this would cause greater variation in gene expression. Further investigation of these 508 variants led to the discovery of 256 examples of epistasis, and 57 of these were replicated in samples from another cohort. Brown et al. calculated that these epistatic interactions explained up to 16% of the variation in gene expression. Furthermore, as well as being involved in epistatic interactions, about 70% of the genetic variants that had an effect on the variation in gene expression were also involved in interactions between genes and the environment.

In addition to showing that epistasis contributes to variation in human traits, the work of Brown et al. could help to uncover interactions behind complex traits—beyond the expression level of a gene—that could not previously be investigated.

effect of genetic variants was dependent on the genetic background, and found frequent evidence of genetic interactions between one or more variants and the originally associated SNPs. Annotating these interacting SNPs to genes revealed common networks of highly connected genes across both populations. In a study of sources of variation in yeast crosses, *Bloom et al. (2013)* carried out a scan for epistasis which discovered 78 pairs of loci where the effect of one was dependent on the genotype of the other, affecting 24 traits. In most cases these interactions explained little of the genetic variation in trait, the median was 3%, but in one case 14% of this variance was explained. Significant interactions between variants have also been seen to affect rice yields (*Huang et al., 2014*) and metabolic traits in yeast (*Wentzell et al., 2007*). An extended recent review of study designs appropriate to detect epistasis in model organisms, and the evidence thus far collected, can be found in *Mackay (2014)*.

However, epistasis has proved harder to identify in human genome-wide association studies. In particular, with classical complex traits there has not been evidence of epistasis on the scale seen in model organisms. This may be in part because of the large number of possible interactions to test in the human genome, and possibly because the genetic architecture is different in a homogeneous outbred population from that of a cross between inbred lines.

*Paré et al. (2010)* have described how an interaction, either genetic or environmental, can induce genotype dependent variance in phenotypes. This effect can be observed without directly modeling the interacting factor. They suggested that SNPs which showed such effects on variance could be prioritized in the search for interactions. We see an example of why this could be true in *Figure 1A*: carriers of C allele of SNP rs230273 show reduced expression when also carriers of the G allele of SNP rs3131691. For carriers of this G allele, this induces a bimodality in expression which appears as a large variance in expression. For those with AA genotype at rs3131691, expression appears independent of

rs230273 genotype; in the absence of the induced bimodality, the variance within this group is much reduced. The interactions causing genotype dependent variance could be with another genetic variant (epistasis, as in our example and the focus of this paper) or an environmental factor.

We therefore adopt the following two step strategy for uncovering epistasis affecting gene expression. We search for: (1) SNPs affecting the variance of expression (v-eQTL) within the 2 Mbp region around the transcription start site (TSS) of the gene, and then (2) SNPs in epistasis with these v-eQTL. Previous work that looked for variance QTL for height and BMI in ~150,000 samples identified one replicated locus (*Yang et al., 2012*). *Wang et al. (2014)* also looked at v-eQTL in gene expression in the same cohort as presented here, where expression was quantified using microarrays rather than sequence based technology (*Grundberg et al., 2012*). They concluded that v-eQTL can often be induced by partial linkage disequilibrium with eQTL. They also discovered differences in expression between monozygotic twins which were dependent on genotype of the twin pair, such differences cannot be induced by these partial linkages and thus point to a gene–environment interaction. The haplotype effect explanation for v-eQTL, combined with a literature which has concluded in many cases epistasis does not contribute to variation in complex traits (*Hill et al., 2008*), led them to conclude epistasis is not a cause of v-eQTL. However, they do not search for examples of epistasis; we do so in this paper, explicitly ruling out haplotype effects. We note that microarray data are also less suitable than RNA-seq for the purpose of detecting v-eQTL, because saturation of signal limits discrimination at extremes (*Wang et al., 2009*). In neither *Yang et al. (2012)* nor *Wang et al. (2014)* were variance QTL directly used to identify epistatic or GxE interactions.

Two papers have also looked at producing a phenotype related to variance, in both cases using the coefficient of variance (CV) within inbred lines to map variants which control the stochastic influence in phenotypic variation (*Ansel et al., 2008*; *Jimenez-Gomez et al., 2011*). In single cell work, and animal models where the environment can be strictly controlled, variance within inbred lines could be seen as stochastic. But we focus our work on where genotype dependent variance is the consequence of a hidden factor, in our case the presence of an interaction between genetic variants, rather than examples where the observations are due to differences in random processes.

There are two other mechanisms by which genotype dependent variance can be induced. Firstly, as *Sun et al. (2013)* have described, standard eQTL working on mean gene expression levels can be mistaken for having variance effects in the presence of a mean–variance relationship. With RNA-seq data, the relationship between mean and variance is clear; as RNA-seq reads are sampled from a Poisson distribution, a square root transformation breaks this link. Secondly, as discussed by the *Wang et al. (2014)* paper described above, haplotype effects can appear as v-eQTL. For example, the situation where a recent strong eQTL co-segregates with a more common SNP (i.e., the SNP is in low $R^2$ with the eQTL, but high D′) could be observed as variance effects of a single SNP. This could also by mistaken for epistasis between two variants which jointly tag the eQTL. We control for this possibility by explicitly considering all possible explanatory eQTL in the full sequence data available for our replication sample.

## Results

We searched for v-eQTL in a dataset of 765 LCL samples from female Caucasian adult twins in the TwinsUK cohort, including 134 monozygotic (MZ) twin pairs and 192 dizygotic (DZ) pairs. The same samples from this cohort have previously been used for eQTL analysis, with expression quantified using microarrays (*Grundberg et al., 2012*). The level of expression of 13,660 genes was determined using whole transcriptome sequencing (RNA-seq). Using a non-parametric association test between SNPs within a cis window of ±1 Mbp around the TSS and the square of the residuals ('Materials and methods'), we identified 497 SNPs as peak v-eQTL for 508 genes (false discovery rate (FDR) <0.05, *Figure 1—figure supplement 1*; *Supplementary file 1A*), 23 reaching Bonferroni significance (nominal p-value <8.9 × 10$^{-10}$). Many of the FDR defined v-eQTL cluster close to the TSS (9.3% are within 10 kb) but they are found at all positions in the window (*Figure 1B*). Of the 508 v-eQTL, 181 are also significant eQTL at a false discovery rate (FDR) of 0.05 (*Figure 1—figure supplement 2*).

To search for epistasis, we scanned the cis windows for a second variant statistically interacting with each of the peak v-eQTL. A forward stepwise analysis identified independent examples of epistasis, not induced by linkage disequilibrium; a statistical test was applied to remove signals related to dominance ('Materials and methods'). This identified 256 independent SNPs in apparent epistasis with the peak v-eQTL for 173 genes (Bonferroni, p-value <1.98 × 10$^{-8}$; *Supplementary file 1B*). To call these signals as genuine genetic interactions we required two further criteria: (i) significant replication

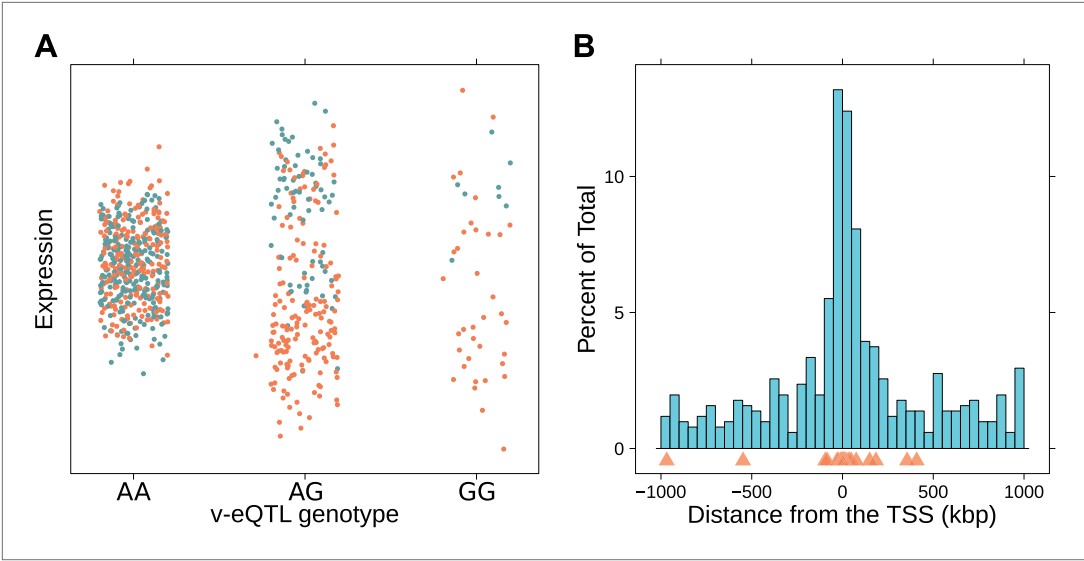

**Figure 1**. Genotype dependent variance analysis identifies candidate SNPs for interactions. These SNPs cluster close to the transcription start site. (**A**) The plot shows expression of the gene *TRIT1*, broken down by v-eQTL genotype (rs3131691), to illustrate how an interaction can be observed as an increase in variance. The genotype at rs3131691 interacts with the genotype of rs230273. Orange individuals are carriers of the C allele at rs230273, which decreases expression only in the AG and GG genotype groups of rs3131691. Observing only expression conditioned on rs3131691, this induced bimodality increases the variance of the observations within these groups. Jitter has been introduced in the x axis to reduce overplotting. (**B**) Histogram of distance from transcription start site in kilobases for the 508 peak v-eQTL hits. Figure shows the clustering of the 508 v-eQTL discovered in the TwinsUK cohort around the transcription start site, with downstream of the TSS counted as positive. The orange triangles below mark the positions of the 26 v-eQTL which replicated in the GEUVADIS cohort.

The following figure supplements are available for figure 1:

**Figure supplement 1**. Peak v-eQTL signals for 13,660 genes.

**Figure supplement 2**. −log10 p value for v-eQTL against−log10 p value for eQTL for 508 v-eQTL hits estimated in the TwinsUK cohort.

**Figure supplement 3**. Variance of expression of ENSG00000164978 (*NUDT2*) is dependent on genotype dosage of rs10972055.

**Figure supplement 4**. Variance of expression of ENSG00000105499 (*PLA2GC4*) is dependent on genotype dosage of rs8109684.

**Figure supplement 5**. Variance of expression of ENSG00000043514 (*TRIT1*) is dependent on genotype dosage of rs3131691.

**Figure supplement 6**. Variance of expression of ENSG00000075234 (*TTC38*) is dependent on genotype dosage of rs6008743.

**Figure supplement 7**. Variance of expression of ENSG00000164111 (*ANXA5*) is dependent on genotype dosage of rs6857766.

**Figure supplement 8**. Variance of expression of ENSG00000137054 (*POLR1E*) is dependent on genotype dosage of rs7033474.

**Figure supplement 9**. Variance of expression of ENSG00000168765 (*GSTM4*) is dependent on genotype dosage of rs542338.

*Figure 1. Continued on next page*

*Figure 1. Continued*
**Figure supplement 10**. Variance of expression of ENSG00000232629 (*HLA-DQB2*) is dependent on genotype dosage of rs114183935.
**Figure supplement 11**. Variance of expression of ENSG00000196735 (*HLA-DQA1*) is dependent on genotype dosage of rs9276807.
**Figure supplement 12**. Variance of expression of ENSG00000160284 (*C21orf56*) is dependent on genotype dosage of rs16978976.

in an independent dataset, and (ii) that the interaction could not be explained by the effect of a third, possibly rare, variant effecting expression as discussed above.

We replicated our scan for v-eQTL and epistatic interactions in 462 samples with LCL RNA-seq data from 1000 Genomes samples collected by the GEUVADIS consortium (*Lappalainen et al., 2013*). *Table 1* reports the results of replication for v-eQTL and epistasis using both FDR and Bonferroni correction for threshold determination. For the 23 v-eQTL that are significant using the Bonferroni threshold, 16 are significant in the GEUVADIS cohort (FDR <0.05), 15 with same direction of effect. Of the 508 v-eQTL, 28 replicated with an FDR <0.05, 26 with same direction of effect. The ten most significant v-eQTL in the GEUVADIS cohort, with matching direction of effect across the two cohorts, are shown in *Figure 1—figure supplements 3–12*.

Of the 256 epistasis associations, information on both the SNP and the gene was available for 246 in the GEUVADIS data. We found that 137 replicated with FDR <0.05, 131 of which had the same direction of effect (*Supplementary file 1B*). p-value enrichment analysis (*Storey, 2002*) indicated that there was replication evidence for 71% of the 246. Moreover, we observed a correlation of 0.58 between the effect sizes of the interactions in both datasets (p-value = $5.9 \times 10^{-24}$), with 202 of the 246 interactions sharing the same direction of effect (p-value = $2.2 \times 10^{-25}$) (*Figure 2—figure supplements 1, 2*).

As discussed in the introduction, it is possible that an observed statistical interaction between two SNPs can be caused by a single true eQTL in linkage disequilibrium with them. For example, a particular combination of alleles across the pair of SNPs could tag a rare causative eQTL. To rule out this possibility, we took advantage of the full sequence for the GEUVADIS replication samples obtained by the 1000 Genomes Project (*The 1000 Genomes Project Consortium, 2012*). For the 131 replicated examples of epistasis we identified all eQTL for the relevant genes amongst all sequenced cis SNPs or indels (a forward stepwise scan identified all eQTL significant with $p<10^{-5}$, 'Materials and methods'). The aim was for good characterisation of eQTL down to low frequency variants, though this is complicated by power and poorer imputation accuracy at such frequencies. We then tested whether the epistatic interaction was still significant in models incorporating each eQTL individually at the same threshold as previously applied. Fifty seven epistasis signals remain significant. *Figure 2A* shows the effect of the epistasis SNP broken down by genotype group on expression of *TRIT1*, *Table 2* and *Figure 2—figure supplements 3–12* report the 10 most significant examples of epistasis in the GEUVADIS cohort, a full

**Table 1.** Replication analysis

| Test | Threshold | Associations (available for testing in GEUVADIS) | Replicate, FDR <0.05 (% success) | Same direction of effect (% success) | π1 |
|---|---|---|---|---|---|
| v-eQTL | FDR <0.05 | 508 (485) | 28 (5.8%) | 26 (93%) | 0.30 |
| v-eQTL | Bonf <0.05 | 23 (23) | 16 (70%) | 15 (94%) | 0.72 |
| Epistasis | Bonf <0.05 | 256 (246) | 137 (56%) | 131 (96%) | 0.71 |

Significant associations (at FDR and Bonferroni thresholds) from the TwinsUK sample were replicated in GEUVADIS samples. The number of overlapping SNPs and genes in both datasets per analysis is shown, as well as the percentage of replicated associations. $\pi_1$ is an estimate of the proportion of replicating loci in the GEUVADIS cohort (*Storey, 2002*).

list is in *Supplementary file 1B*. For all plotted interactions, the direction of effect was consistent within v-eQTL genotype groups across cohorts. In at least two instances we see sign epistasis, the effect of one SNP reverses direction conditional on the other SNP (*Figure 2—figure supplements 7, 9*).

We estimated the proportion of variance explained by the interaction in the GEUVADIS cohort to avoid over-estimating effects because of winner's curse. As a result, we were able to determine that up to 16% of the variance in gene expression was explained by considering the interaction between the variants, with an average additional variance explained of 4.3% (*Table 2*; *Supplementary file 1B*; *Figure 3*). For the eight genes for which we replicated independent interactions with the v-eQTL, we found that in total up to 10.4% of the variance was explained by these multiple interactions, with an average of 5.1%. For 24 out of 57 the replicated examples of epistasis, the interaction explains more variance than the additive effects of the SNPs. We show as an example the gene *TRIT1* (*Figure 2*). The v-eQTL (rs3131691) for *TRIT1* lies on the boundary of an ENCODE defined LCL weak enhancer (*Dunham et al., 2012*; *Rosenbloom et al., 2013*) upstream of the gene, while the SNP in epistasis (rs230273) lies on the boundary of a downstream LCL enhancer region (*Figure 2B*). The v-eQTL is also 28 bp upstream of a strong eQTL signal (rs34387655). This eQTL has minor allele frequency (MAF) 0.08, and is in high D' with the v-eQTL (MAF = 0.30), suggesting that the eQTL could be a recent mutation co-segregating with one allele of the v-eQTL. But this eQTL cannot explain the observed interaction, which was still significant when analyzing only major allele homozygotes for the eQTL (p-value = 0.0095). Therefore, we conclude that two causal loci act on the weak enhancer in two different ways; rs34387655 has a direct effect on the enhancer while rs3131691 acts in conjunction with the epistasis variant rs230273 (or variants in linkage disequilibrium with these SNPs act in these ways).

The discussion up to this point concerns SNPs in cis with the expressed gene. Looking for examples of trans SNPs (>5 Mbp from the TSS) in epistasis with the v-eQTL yielded no hits that replicated in the GEUVADIS cohort. However, using the twin design we were able to address the contribution of long range epistasis by a heritability analysis. Assuming no recombination in the cis region, the proportion of the cis window that dizygotic twins (DZ) inherited identically by descent is either 0, 0.5 or 1 and this allows us to perform a linkage analysis to estimate the proportion of variance explained by variants in the cis region, the trans region (5 Mbp away from the TSS) and interactions between the two. We had information about the IBD sharing around 273 of the 508 v-eQTL genes. For 15 of these, interactions between the cis and trans regions explain more than 10% of the variance in expression. For all of these there is greater evidence of cis-trans epistasis affecting expression than an influence of common environment, and for 9 of the 15 the interaction effect was more than the estimated combined direct genetic contribution of both cis and trans variants (*Supplementary file 1C*).

The presence of v-eQTL can be induced by gene–environment interactions, as well as epistasis or haplotype effects. Because our data come from a twin cohort, which includes monozygotic (MZ) twin pairs, we have another measure of variability within the dataset: the discordance in expression between MZ twins. Genotype dependent differences in expression within MZ pairs cannot be induced by epistasis or haplotype effects, as both twins share the same genetic background. Therefore, evidence that v-eQTL are also discordant eQTL (d-eQTL) would suggest that v-eQTL could also have a GxE explanation, including possibly interactions between the genome and the epigenome (*Martin et al., 1983*; *Reynolds et al., 2007*; *Figure 4A*). Using our MZ data, we have tested our 508 v-eQTL for evidence that they are also d-eQTL; using the methods from *Storey (2002)* we estimate that 70% of the v-eQTL act in this manner. This suggests that GxE interactions are common amongst these variants ('Materials and methods', *Figure 4B*; *Supplementary file 1A*). In total, 176 of the 508 v-eQTL show significant effects on discordance (FDR <0.05). Of these 176, we estimate the proportion that are also eQTL as 40.3%, less than the proportion of all v-eQTL which act as eQTL.

By looking at variance between individuals and discordance between monozygotic twins, we mirror an approach which looked at robustness of phenotypes to genetic and environmental influences (*Fraser and Schadt, 2010*). In this study of gene expression traits, differences between inbred mouse strains were called 'genetic robustness QTL' (GR-QTL). These correspond to our definition of v-eQTL, and the paper discusses how they can be induced by epistatic interactions. The paper also looks at QTL for within strain variance, analogous to our d-eQTL and referred to as 'environmental robustness QTL' (ER-QTL), and describe them as induced by gene–environment interactions. They reported finding both GR-QTL and ER-QTL in mice, *Arabidopsis* and *S. cerevisiae*.

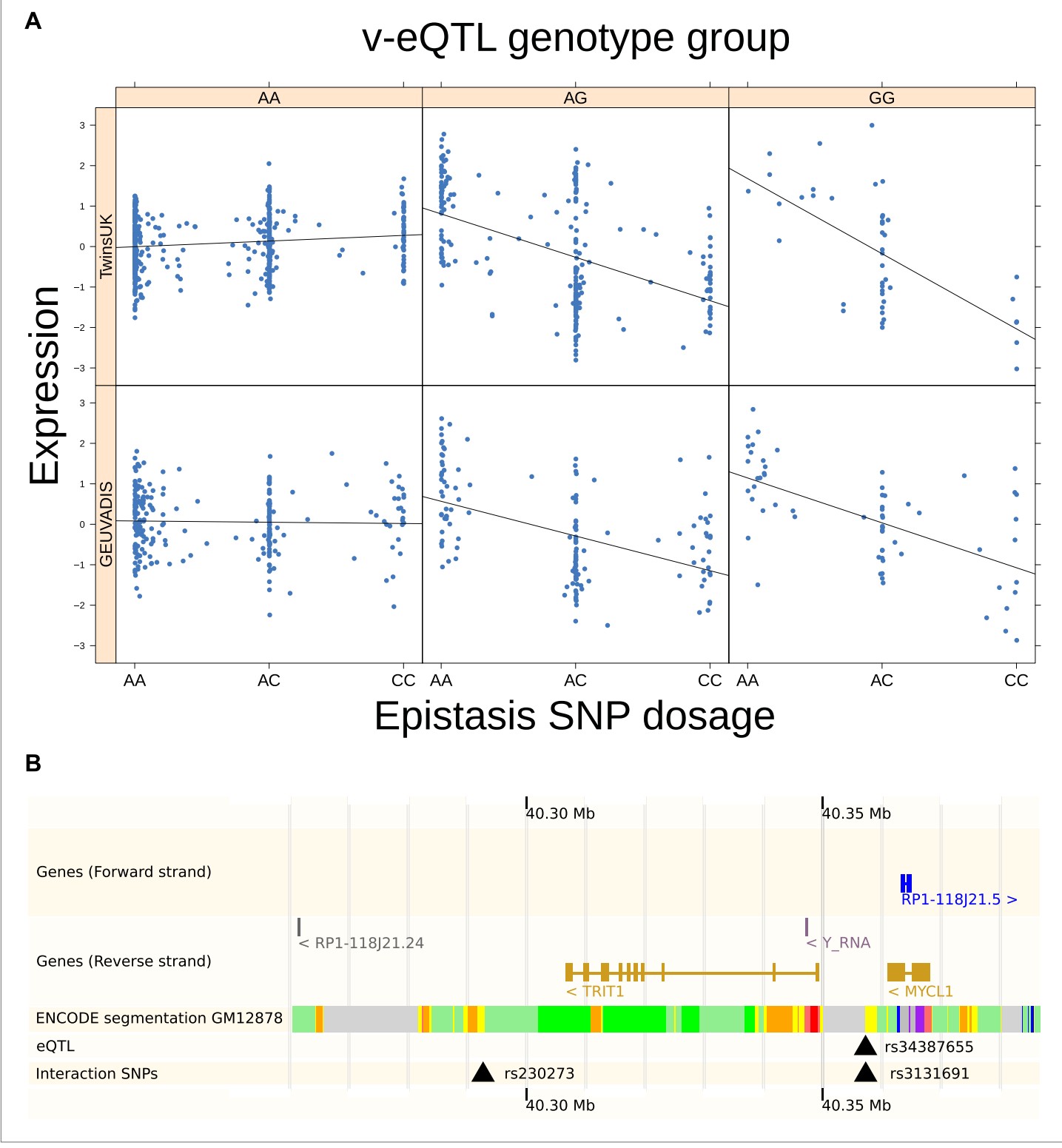

**Figure 2**. *TRIT1* expression is affected by an interaction between two SNPs, lying on the boundaries of two separate enhancer regions, in both TwinsUK and GEUVADIS cohorts. (**A**) Expression of *TRIT1* is shown, with a separate panel for each v-eQTL (rs3131691) genotype group. Relationship between expression and imputed genotype dosage of the epistasis SNP (rs230273) is shown to be conditional on v-eQTL genotype. Expression from TwinsUK individuals is shown in the upper panels, GEUVADIS individuals in the lower panels. Best fit lines show different SNP effects for the epistatic SNPs in different v-eQTL genotype groups, these lines are constructed ignoring twin structure in the case of the TwinsUK sample and population in the

*Figure 2. Continued on next page*

*Figure 2. Continued*

GEUVADIS cohort. (**B**) SNPs affecting *TRIT1* expression are near regulatory elements. Position of v-eQTL (rs3131691), interacting epistasis SNP (rs230273) and a nearby eQTL (rs34387655) affecting *TRIT1* expression are shown. ENCODE segmentation analysis shows regulatory elements around *TRIT1* (reverse strand gene). Colours indicating regions are: yellow = weak enhancer, orange = strong enhancer, red = strong promoter, light red = weak promoter, purple = poised promoter, dark green = transcriptional transition/elongation, light green = weakly transcribed, blue = insulator, and light grey = heterochromatin or repetitive/copy number variation.

The following figure supplements are available for figure 2:

**Figure supplement 1**. Evidence for epistasis in twins against evidence for epistasis in 1000 Genomes for the 246 significant hits.

**Figure supplement 2**. Estimate of interaction effect size in 1000 Genomes and twins cohorts.

**Figure supplement 3**. ENSG00000164978 (*NUDT2*) expression is affected by an interaction between two SNPs in both TwinsUK and GEUVADIS cohorts.

**Figure supplement 4**. ENSG00000232629 (*HLA-DQB2*) expression is affected by an interaction between two SNPs in both TwinsUK and GEUVADIS cohorts.

**Figure supplement 5**. ENSG00000232629 (*HLA-DQB2*) expression is affected by an interaction between two SNPs in both TwinsUK and GEUVADIS cohorts.

**Figure supplement 6**. ENSG00000006282 (*SPATA20*) expression is affected by an interaction between two SNPs in both TwinsUK and GEUVADIS cohorts.

**Figure supplement 7**. ENSG00000204531 (*POU5F1*) expression is affected by an interaction between two SNPs in both TwinsUK and GEUVADIS cohorts.

**Figure supplement 8**. ENSG00000021355 (*SERPINB1*) expression is affected by an interaction between two SNPs in both TwinsUK and GEUVADIS cohorts.

**Figure supplement 9**. ENSG00000164111 (*ANXA5*) expression is affected by an interaction between two SNPs in both TwinsUK and GEUVADIS cohorts.

**Figure supplement 10**. ENSG00000137310 (*TCF19*) expression is affected by an interaction between two SNPs in both TwinsUK and GEUVADIS cohorts.

**Figure supplement 11**. ENSG00000204525 (*HLA-C*) expression is affected by an interaction between two SNPs in both TwinsUK and GEUVADIS cohorts.

**Figure supplement 12**. ENSG00000176531 (PHLDB3) expression is affected by an interaction between two SNPs in both TwinsUK and GEUVADIS cohorts.

**Figure supplement 13**. The distance in kilobases from the 246 variants in epistasis to the v-eQTL, plotted against the –log10 p value in 1000 Genomes sample.

## Discussion

The importance of non-additive variation to explaining missing heritability has been much debated (*Hill et al., 2008*; *Zuk et al., 2012*). Here, we were able to report specific examples of interactions explaining noticeable fractions of variation in human gene expression, with in many cases the interaction contributing more than the marginal effects to overall variance. Estimating variance components from pedigrees and twin model studies has concentrated on additive variance, to estimate the narrow sense heritability. The assumption has been that resemblance between related individuals is determined chiefly by additive variation (*Falconer and Mackay, 1996*). An overview of analyses of many phenotypes in many organisms concluded that there was little evidence for non-additive variation playing a large role in phenotypic variation (*Hill et al., 2008*). Indeed, the authors provided a theoretical argument that the total contribution of all interacting loci to variance is well approximated by their additive contribution, when the allele frequencies are as predicted by the neutral model. The analysis presented here is powered chiefly to discover common interacting variants, however the result on the neutral model implies there may be many more examples of epistasis which are not statistically detectable without very large samples.

**Table 2.** Effect size estimates and significance for the ten most significant replicated interactions in TwinsUK and GEUVADIS

| Gene | Chr | v-eQTL | Interacting epistasis SNP | Interaction variance in TwinsUK | Interaction variance in GEUVADIS | Additive variation in GEUVADIS | p-value in TwinsUK | p-value in GEUVADIS |
|---|---|---|---|---|---|---|---|---|
| NUDT2 | 9 | rs10972055 | rs10814083 | −0.328 | −0.128 | 0.310 | $1.88 \times 10^{-53}$ | $5.43 \times 10^{-22}$ |
| HLA-DQB2 | 6 | rs114183935 | rs1049130 | −0.337 | −0.161 | 0.099 | $1.83 \times 10^{-62}$ | $2.91 \times 10^{-21}$ |
| HLA-DQB2 | 6 | rs114183935 | rs9274666 | −0.368 | −0.119 | 0.158 | $3.45 \times 10^{-18}$ | $1.04 \times 10^{-16}$ |
| SPATA20 | 17 | rs12943759 | rs1122634 | 0.301 | 0.078 | 0.404 | $3.12 \times 10^{-69}$ | $1.42 \times 10^{-15}$ |
| POU5F1 | 6 | rs116627368 | rs115631087 | 0.311 | 0.116 | 0.008 | $6.95 \times 10^{-34}$ | $6.63 \times 10^{-14}$ |
| SERPINB1 | 6 | rs318452 | rs6940344 | −0.227 | −0.102 | 0.117 | $2.40 \times 10^{-36}$ | $7.66 \times 10^{-14}$ |
| ANXA5 | 4 | rs6857766 | rs12511956 | −0.411 | −0.104 | 0.056 | $3.09 \times 10^{-37}$ | $3.81 \times 10^{-13}$ |
| TCF19 | 6 | rs115523621 | rs115921994 | −0.585 | −0.076 | 0.201 | $2.59 \times 10^{-36}$ | $1.48 \times 10^{-11}$ |
| HLA-C | 6 | rs114916097 | rs116012228 | 0.160 | 0.077 | 0.183 | $3.35 \times 10^{-18}$ | $2.17 \times 10^{-11}$ |
| PHLDB3 | 19 | rs10409591 | rs2682547 | −0.270 | −0.0858 | 0.0569 | $1.67 \times 10^{-14}$ | $4.83 \times 10^{-11}$ |

Effect sizes are reported as the proportion of variance explained by the interaction. Sign of effect size reflects direction of interaction effect: positive implies combined effect of the alternate alleles is an increase in expression greater than predicted by separate additive effects, and negative that it is less.

Specifically in gene expression, progress has recently been made to move beyond a solely additive view of variation. *Becker et al. (2012)* produced evidence for the existence of cis-trans epistasis, though they do not report individual examples which were significant when controlling for all tests and did not consider the contribution of these interactions to phenotypic variation. Further work from *Powell et al. (2013)* looked to dissect the phenotypes into dominant and additive components. As with our dissection of cis-trans epistasis, additive genetic variation was most consistently observed, though 960 probes had a dominant component to variation; for a subset of these a non-additive eQTL was proposed. All in all, these global results together with the replicated epistatic interactions presented here suggest a moderate influence of non-additive genetic effects on gene transcription variation.

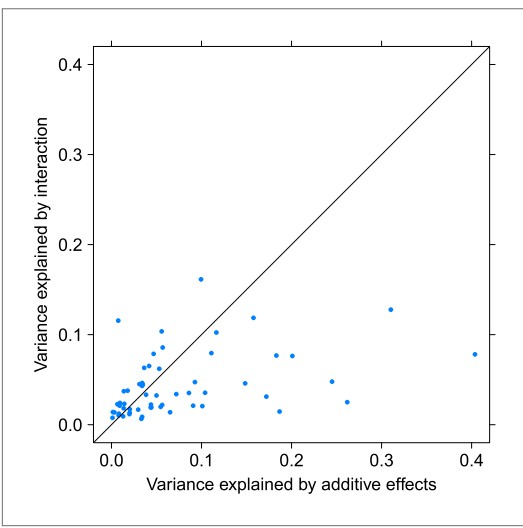

**Figure 3**. Variance explained by additive and interacting variants for 57 replicated examples of epistasis in the GEUVADIS cohort. We show the variation explained by the interaction of two SNPs on phenotype, compared to the additive contribution of the SNPs.

The majority of the interactions are close to each other and to the TSS (*Figure 2—figure supplement 13*), consistent with a direct molecular interaction. However, despite physical proximity they are, because of the statistical discovery strategy, in low linkage disequilibrium. There has been discussion in the literature about how interactions between variants affecting fitness can change the linkage disequilibrium structure of a region, by bringing variants which alter the local recombination rate under indirect selection (*Otto and Feldman, 1997*). In the case of positive epistasis, where the combined effect on fitness of the deleterious alleles is mitigated by their joint contribution, selection would favour a decrease in the recombination rate between the loci. This was seen in *Lappalainen et al. (2011)*: non-synonymous, possibly deleterious, coding mutations together with an eQTL which adjusts expression would be an example of positive epistasis. In support of the theoretical result, such variants were frequently observed in high linkage disequilibrium in their results. In contrast, the approach we take here requires linkage disequilibrium to have broken

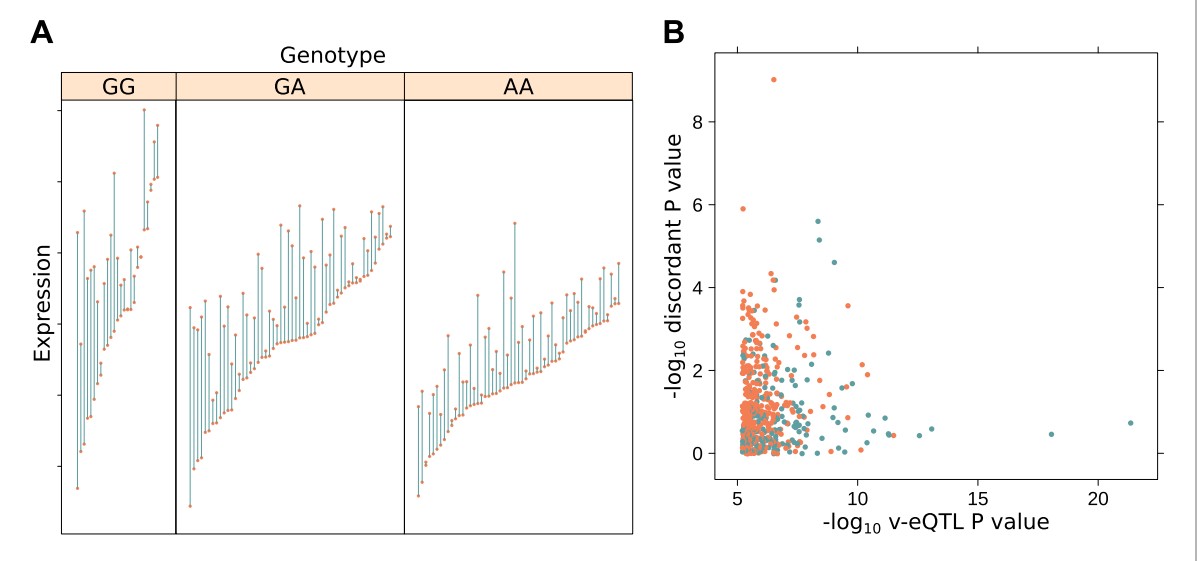

**Figure 4**. Increased discordance within MZ twin pairs identifies GxE interactions. (**A**) We show discordance in expression between MZ twin pairs for the gene *BAMBI* broken down by v-eQTL genotype (rs10826519). Discordance is greatest in the GG genotype group (mean difference between MZ twins is 1.12), decreasing with each additional copy of the A allele (mean discordance is 0.85 for GA genotype group, 0.60 for AA). Since MZ twins are genetically identical, genotype dependent discordance in expression must be a consequence of environment, pointing to GxE. We observe that the SNP also has an effect on the mean level of expression ($p = 5.42 \times 10^{-19}$). (**B**) $-\log10$ p values for genotype dependent discordance in MZ twins against $-\log10$ p values for peak v-eQTL. The blue dots represent points where there is a significant epistasis hit with the v-eQTL, orange where no such interaction was detected. For many of the strong v-eQTL with little evidence of discordance we can identify an epistatic interaction which explains the increase in variance. However, for some loci with strong evidence of genotype dependent MZ discordance we also detect an epistatic interaction, suggesting both epistasis and GxE acts on these genes.

down between variants in order to distinguish an interaction between two variants from a dominant effect of a single locus. As a consequence, we are powered more to detect epistasis which amplifies the effect of deleterious mutations, rather than positive epistasis as described by *Lappalainen et al. (2011)*. Therefore, examples of epistasis of the type they describe would be missed by our methodology (indeed, the five non-synonymous SNPs we discover to be involved in interactions in the TwinsUK dataset are all predicted by PolyPhen score to be benign with the exception of a one (rs150369207) which is classed as possibly damaging for only one out of nine coding transcripts).

A recent paper has also looked for evidence of epistasis affecting transcription in humans (*Hemani et al., 2014*), using array expression from whole blood and searching the entire space of all possible pairwise interactions. They discover 501 interactions, affecting expression of 238 genes in 846 samples, and replicate 30 examples in an independent dataset at Bonferroni significance level. The interactions discovered are chiefly cis-trans; of the 501 there are 26 cis–cis interactions and 13 trans–trans. The apparent lower replication rate compared to our study may reflect the greater success that has been seen replicating cis effects than trans effects for standard eQTL (*Grundberg et al., 2012*). *Grundberg et al. (2012)* also reported that LCLs (the tissue used in our study) showed stronger genetic effects compared to environmental contribution than seen in primary tissues. Finally, RNA-seq has been shown as a more reliable phenotype than array based measures (*Marioni et al., 2008*). We believe all these factors contribute to our success rate in replicating epistatic interactions.

In conclusion, we report 26 replicated variance eQTL and 57 replicated cis epistatic interactions, which explain up to 16% of the variance of our phenotypes. In almost a half of cases, more variance is explained by the interaction than by single additive effects. Furthermore, we have also shown substantial evidence for gene by environment interactions. We have shown that a proportion of variation of molecular phenotypes can be ascribed to genetic interactions, and that v-eQTL are a valid way of discovering them. Densely phenotyped cohorts are now commonly collecting such molecular data, and therefore there is considerable scope to look both for more of this type of interactions, and for the

particular environments involved in GxE. The ability to find genetic interactions affecting molecular phenotypes also suggests a hypothesis driven path by which genetic interactions underlying more complex traits may be identified.

# Materials and methods

## Genotying and imputation

Samples were genotyped on a combination of the HumanHap300, HumanHap610Q, 1 M-Duo and 1.2MDuo 1M Illunnia arrays. Samples were pre-phased using IMPUTE2 (*Howie et al., 2009*) with no reference panel, then imputed into the 1000 Genomes Phase 1 reference panel (interim, data freeze, 10 November 2010, *The 1000 Genomes Project Consortium 2012*). Post imputation, SNPs were removed if MAF <0.01 or IMPUTE info value <0.8.

## RNA processing

Samples were prepared for sequencing with the Illumina TruSeq sample preparation kit (Illumina, San Diego, CA) according to manufacturer's instructions and were sequenced on a HiSeq2000 machine. Afterwards, the 49-bp sequenced paired-end reads were mapped to the GRCh37 reference genome (*The International Human Genome Sequencing Consortium, 2001*) with BWA v0.5.9 (*Li and Durbin, 2009*). We use genes defined as protein coding in the GENCODE 10 annotation (*Harrow et al., 2012*), removing genes with more than 10% zero read count. RPKM values were root mean transformed. PEER software (*Parts et al., 2011*) was used to remove 50 latent factors; age and body mass index were included when factors were constructed, to prevent removal of important environmental factors. Data were then quantile normalised.

## v-eQTL

GRAMMAR (*Aulchenko et al., 2007*) was used to remove correlations between related individuals. Expression of each gene was tested against every SNP within 1 Mbp of the TSS. First, any eQTL effects were removed by regressing expression on the posterior probability of being a heterozygote and the posterior probability of being a minor allele homozygote. The residuals were squared, giving a measure of distance from the mean expression of that genotype class for all individuals. A Spearman rank correlation test between this 'distance' and genotype dosage was used to assess evidence of variance effects. A set of five permutations, consistent across all tests to consider linkage disequilibrium structure between SNPs, was applied to the distance residuals and the spearman correlation test was applied as before to estimate the distribution of the test statistic under the complete null hypothesis of no variance effects. An FDR was calculated as the proportion of permuted statistics more significant, divided by 5. This two stage procedure where relatedness was regressed out separately from v-eQTL mapping was adopted to make the full scan for v-eQTL computationally feasible.

## Epistasis

The R package lme4 (*Bolker, 2013*) was used to fit linear mixed models using maximum likelihood to model expression as a function of genetic interactions. The models, with a full description of how the twin structure is captured, are presented in the section 'Equations'. A forward stepwise scheme, as used in *Lappalainen et al. (2013)* to map standard eQTL, was used to discover independent examples of epistasis. Assuming the K-1 significant examples of epistasis had been discovered, a complete scan of every SNP in the cis window tested for evidence of epistasis with the v-eQTL (using a likelihood ratio test of *Equation 2* nested into *Equation 1*, testing the hypothesis $c_K = 0$), conditioned on all previously discovered interactions. If the most significant SNP was Bonferroni significant (p<1.98 × 10$^{-8}$), the SNP was added to the list and the process continued, otherwise the list was considered complete. This revealed 275 examples of epistasis, affecting expression of 178 genes. To exclude the possibility that significant interactions could be explained by a non-additive genetic effect of the original v-eQTL appearing as epistasis between the v-eQTL and another variant in tight linkage disequilibrium, a further conditional analysis tested the epistasis term conditional on the model it was discovered in and a non-additive effect of the v-eQTL (testing nested models, *Equation 3* and *Equation 4* for $c_K = 0$). SNPs which were not Bonferroni significant at the same threshold (p<1.98 × 10$^{-8}$) were removed, leaving 256 epistatic interactions affecting 173 genes. Proportion of variance for linear mixed models was calculated as described in *Nakagawa and Schielzeth (2012)*. Scripts to analyse the data are provided in Supplementary material.

## Equations

Denoting individual $i$, expression by $y_i$, dosage of v-eQTL by $S_{iv}$, dosage of the kth discovered epistatic SNPs by $S_{ik}$, probability that the v-eQTL is a heterozygote by $S_{iv}^{het}$, and the probability that the v-eQTL is a minor allele homozygote by $S_{iv}^{hom}$, we have modelled expression in the following ways:

$$y_i = \mu + aS_{iv} + \sum_{k=1}^{K-1}\left(b_k S_{ik} + c_k S_{iv} S_{ik}\right) + b_K S_{iK} \quad + \beta_i + \gamma_i + \epsilon_i \tag{1}$$

$$y_i = \mu + aS_{iv} + \sum_{k=1}^{K-1}\left(b_k S_{ik} + c_k S_{iv} S_{ik}\right) + b_K S_{iK} + c_K S_{iv} S_{iK} \quad + \beta_i + \gamma_i + \epsilon_i \tag{2}$$

$$y_i = \mu + a^{het}S_{iv}^{het} + a^{hom}S_{iv}^{hom} + \sum_{k=1}^{K-1}\left(b_k S_{ik} + c_k S_{iv} S_{ik}\right) + b_K S_{iK} \quad + \beta_i + \gamma_i + \epsilon_i \tag{3}$$

$$y_i = \mu + a^{het}S_{iv}^{het} + a^{hom}S_{iv}^{hom} + \sum_{k=1}^{K-1}\left(b_k S_{ik} + c_k S_{iv} S_{ik}\right) + b_K S_{iK} + c_K S_{iv} S_{iK} \quad + \beta_i + \gamma_i + \epsilon_i \tag{4}$$

where

$$\beta_i \sim N\left(0, \sigma_{FAM}^2\right)$$

$$\gamma_i \sim N\left(0, \sigma_{MZ}^2\right)$$

$$\epsilon_i \sim N\left(0, \sigma^2\right)$$

To correctly model the twin structure we require that $\beta_i = \beta_j$ when $i$ and $j$ are twins, and $\gamma_i = \gamma_j$ when $i$ and $j$ are MZ twins (capturing the increased genetic correlation of MZ twins).

## Heritability

A variance components model was fitted in the program solar (*Almasy and Blangero, 1998*) where the covariance matrix for the trait is written:

$$\Omega = \Pi_{cis}\sigma_{cis}^2 + \Pi_{trans}\sigma_{trans}^2 + \Pi_{cis-trans}\sigma_{cis-trans}^2 + I\sigma_e^2$$

$\Pi_{cis}$ and $\Pi_{trans}$ are the proportion of cis and trans alleles that twins share inherited identically by descent and $\Pi_{cis-trans}$ is the Hadamard product of these matrices. Parameters were estimated by maximum likelihood and proportion of variance explained by cis-trans interactions was estimated as:

$$\frac{\sigma_{cis-trans}^2}{\sigma_{cis}^2 + \sigma_{trans}^2 + \sigma_{cis-trans}^2 + \sigma_e^2}$$

For comparison, the model without cis-trans interactions but with a common environment term was fitted, and the two models compared using likelihood.

## Discordant QTL

Maximum expression of the two twins was regressed on minimum expression of the twin pair and genotype of the twin pair to detect whether the relationship between max and min expression was conditional on genotype.

## GEUVADIS replication

Raw RPKM values were root transformed, 20 principal component factors were removed and then the data were quantile normalised. Evidence for v-eQTL and epistasis was calculated as before, with indicator variables for study population (CEU, YRI, TSI, GBR, FIN) to control for population effects. Epistasis was assessed for each SNP individually, as LD induced multiple signals and dominance effects had been removed in the TwinsUK sample. To ensure that our results are not caused by heteroskedasticity, we have considered various transformations to remove this issue and found the results to be robust. In particular, of the 131 statistically significant interactions in the GEUVADIS cohort, 126 are also significant when log transformed data is analysed (a typical way of accounting for heteroskedasticity). To eliminate

confounding with eQTL variants, an identical forward stepwise cis eQTL scan to that used in *Lappalainen et al. (2013)* reported all eQTL significant at $p < 10^{-5}$ in the GEUVADIS dataset. A *t* test for each reported eQTL assessed significance of the interaction conditional on the v-eQTL, epistasis SNP and the eQTL. If the greatest p value, over all possible eQTL, did not meet the FDR cut-off the SNP was removed from the list of interactions. FDR was calculated using the qvalue package (*Dabney and Storey, 2014*) in R (*R Development Core Team, 2008*) using the default settings with the exception that lambda was restricted to lie within the range of the p values to prevent overly lenient correction. The replication dataset together with functions to reproduce the results are provided in *Supplementary files 2–4*.

## ENCODE segmentation

Segmentation analysis for LCL cell line GM12878 was downloaded from the UCSC website on 11/6/2013, url: http://hgdownload.cse.ucsc.edu/goldenPath/hg19/encodeDCC/wgEncodeBroadHmm/wgEncodeBroadHmmGm12878HMM.bed.gz.

## Sequence data

Sequence data has been deposited at the European Genome-phenome Archive (EGA, http://www.ebi.ac.uk/ega/) under accession number EGAS00001000805.

## Acknowledgements

The TwinsUK study was funded by the Wellcome Trust; European Community's Seventh Framework Programme (FP7/2007-2013). The study also receives support from the National Institute for Health Research (NIHR)-funded BioResource, Clinical Research Facility and Biomedical Research Centre based at Guy's and St Thomas' NHS Foundation Trust in partnership with King's College London. SNP Genotyping was performed by The Wellcome Trust Sanger Institute and National Eye Institute via NIH/CIDR. Some computation was performed at the Vital-IT centre for high-performance computing of the SIB Swiss Institute of Bioinformatics (http://www.vital-it.ch).

## Additional information

### Competing interests

ETD: Reviewing editor, *eLife*. The other authors declare that no competing interests exist.

### Funding

| Funder | Grant reference number | Author |
| --- | --- | --- |
| Wellcome Trust | WT098051 | Richard Durbin |
| Louis-Jeantet Foundation | | Emmanouil T Dermitzakis |
| National Institutes of Health | | Emmanouil T Dermitzakis, Timothy D Spector |
| Swiss National Science Foundation | | Emmanouil T Dermitzakis |
| European Research Council | | Emmanouil T Dermitzakis, Timothy D Spector |
| Canadian Institutes of Health Research | | Hou-Feng Zheng, J Brent Richards |
| Fonds de Recherche Sante de Quebec | | Hou-Feng Zheng, J Brent Richards |
| Quebec Consortium for Drug Discovery | | Hou-Feng Zheng, J Brent Richards |
| South East Norway Health Authority | 2011060 | Andrew Anand Brown |
| European Union | 259749 | Andrew Anand Brown, Alfonso Buil, Ana Viñuela, Timothy D Spector, Emmanouil T Dermitzakis, Richard Durbin |

The funders had no role in study design, data collection and interpretation, or the decision to submit the work for publication.

## Author contributions

AAB, Conception and design, Analysis and interpretation of data, Drafting or revising the article; AB, AV, TL, Acquisition of data, Drafting or revising the article; KSS, Conception and design, Drafting or revising the article; H-FZ, JBR, Imputed genotype data into 1000 Genomes reference panel, Approved final manuscript; TDS, Conception and design, Acquisition of data; ETD, RD, Conception and design, Acquisition of data, Drafting or revising the article

## Ethics

Human subjects: This project was approved by the ethics committee at St Thomas' Hospital London, where all the biopsies were carried out. Volunteers gave informed consent and signed an approved consent form prior to the biopsy procedure. Volunteers were supplied with an appropriate detailed information sheet regarding the research project and biopsy procedure by post prior to attending for the biopsy. The St Thomas' Research Ethics Committee (REC) approved on 20th September 2007 the protocol for dissemination of data, including DNA, with the REC reference number RE04/015. On 12th of March of 2008, the St Thomas' REC confirmed this approval extends to expression data.

# Additional files

### Supplementary files

• Supplementary file 1. **A**: peak vQTL hits in TwinsUK cohort with evidence of eQTL and discordant QTL and replication evidence in GEUVADIS cohort. **B**: significant epistasis hits in TwinsUK cohort with p values and effect size estimates in GEUVADIS cohort. **C**: contribution of cis variants, trans variants, interactions between the two and unique environment to variation in gene expression.

• Supplementary file 2. R functions applied to data from the TwinsUK cohort to test individual SNPs for variance effects, to map all independent epistatic interactions with the v-eQTL in the cis window and to eliminate dominance effects from list of epistatic interactions.

• Supplementary file 3. R workspace containing replication data from the GEUVADIS cohort (*Lappalainen et al., 2013*) together with functions to repeat the replication analysis.

• Supplementary file 4. Read me file explaining objects present in SM2.

### Major dataset

The following dataset was generated:

| Author(s) | Year | Dataset title | Dataset ID and/or URL | Database, license, and accessibility information |
|---|---|---|---|---|
| Brown AA, Buil A, Viñuela A, Lappalainen T, Zheng HF, Richards JB, Small KS, Spector TD, Dermitzakis ET, Durbin R | 2013 | Eurobats LCL RNA-seq data | EGAS00001000805 | RNA-seq data are being deposited in EBI-EGA (http://www.ebi.ac.uk/ega/) for controlled access, release on publication. The DTR twin register is currently set up as a supported access resource for the research community. All data access requests are overseen by the TwinsUK Resource Executive Committee (TREC). Requests for collection of new or existing data/material should be processed by submitting a completed DTR Data/Material Access Proposal Form (http://www.twinsuk.ac.uk/data-access/submission-procedure/). |

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
