## [Decision Letter]

Thank you for sending your work entitled “Genetic interactions affecting human gene expression identified with variance association mapping” for consideration at *eLife*. Your article has been favourably evaluated by a Senior editor, a Reviewing editor, and 2 peer reviewers.

The only substantive concern is that the paper should be re-written because the concepts and methods need to be better explained for non-specialist readers. In particular, it should be made clearer why showing that two loci (SNPs) contributing non-additively to genotype-specific variance is direct evidence of epistasis. There are also presumably specific assumptions in the models, such as the dependence of variance on scale, the type of interaction, or the complex effects of LD, and these should be made clearer.

In terms of methodology, Step 1, the identification of v-eQTL, does not appear to leverage the twin design (“GRAMMAR was used to remove correlations between individuals”) and this should be explained more clearly. Step 2, “Epistasis” does use the twin structure and is based on a LRT comparing linear mixed models with and without an interaction term. What is the form of the interaction term? There are many ways to encode it which can involve more than one parameter for SNPs not in D'=1. Why use a non-parametric test for v-eQTL discovery and then a LMM for interaction? Although the data are quartile normalised, are the squared residuals and what is the effect of outliers? The conditional analysis presumably includes SNPs one-by-one to check the association holds *–* does imputation uncertainty matter here? Please also clarify why the influence of a second eQTL doesn't have an impact on the result.

In the main text: after identification of v-eQTL “to search for epistasis we scanned the cis windows for a second variant statistically interacting with each of the peak v-eQTL”. It would be helpful to include a mathematical description of the model.

---

## [Author Response]

Many of the comments were about a lack of clarity in the methods and explanation: we have in response expanded the paper and included a more detailed motivation for following our path from variance to epistasis.

In the course of expanding the Methods section and replying to the reviewers we re-examined some of the analysis. In particular, we realised that a forward stepwise procedure based on Bonferroni significance would be preferable to the backwards stepwise algorithm we originally used to remove non-independent signals. There are two reasons for this:

1) The backward procedure we applied looked at whether there was sufficient evidence to remove the alternative hypothesis. A forward stepwise procedure asks whether there is sufficient evidence to reject the null hypothesis, the standard approach in statistical inference.

2) The forward stepwise approach has been commonly applied in the literature, e.g., [22] and Battle et al. (2014).

Compared to the previous approach, which yielded no genes with multiple examples of epistasis, we now have identified 83. That is, we were able to find 83 genes where more than one independent SNP showed evidence of an interaction with the v-eQTL, accounting for LD. Details on the methodology and new results have been included in the manuscript.

While implementing these changes, we also became aware of two coding mistakes made during the analysis. Correcting these has improved our results dramatically. Firstly, we corrected a mistake while converting the GEUVADIS dataset genotype information; in combination with the new approach to detect more than one epistatic interaction, this resulted in substantially more replicated examples of both v-eQTL and epistasis in the GEUVADIS cohort. Secondly, there was a mistake in defining the location of the TSS on the negative strand for the TwinsUK analysis. Within the properly defined cis-window we found 7 new v-eQTL, bringing the total to 508.

Because we were able to replicate more examples of epistasis, we have expanded our discussion of the relative impact of interacting and additive effects on variance, including new figures.

Finally, since we submitted the paper the GEUVADIS consortium have reported their results and made the replication data publicly available. We would therefore like to make the processed replication data available as supplemental data for the paper, in an R dataset which also includes functions which will repeat the analysis. This will allow anyone to easily repeat the analysis and check the methodology. We also make available the R scripts used to analyse the TwinsUK sample to allow the methods applied to this dataset to be inspected. We are in the process of depositing the RNA-seq data in EBI-EGA for controlled access, with release on publication.

Below we address each of the reviewers’ concerns:

*The only substantive concern is that the paper should be re-written because the concepts and methods need to be better explained for non-specialist readers. In particular, it should be made clearer why showing that two loci (SNPs) contributing non-additively to genotype-specific variance is direct evidence of epistasis. There are also presumably specific assumptions in the models, such as the dependence of variance on scale, the type of interaction, or the complex effects of LD, and these should be made clearer*.

We have added two new paragraphs to the Introduction (fourth and seventh), which we hope suitably summarise our motivations and the possible causes of genotype dependent variance, as well as modelling assumptions.

*In terms of methodology, Step 1, the identification of v-eQTL, does not appear to leverage the twin design (“GRAMMAR was used to remove correlations between individuals”) and this should be explained more clearly. Step 2, “Epistasis” does use the twin structure and is based on a LRT comparing linear mixed models with and without an interaction term*.

The justification for using GRAMMAR is purely computational, a full scan of all cis windows for v-eQTL involves ∼65 000 000 tests. Ideally we would like to construct residuals which control out twin structure and general SNP effect simultaneously for every SNP as currently this is assumed to maximise power (as argued in Zhou and Stephens (2012)). However, this is computationally infeasible. Instead we adopted a two stage procedure, the twin structure is removed from the phenotype, then each SNP effect can be removed separately using a much faster linear model. The epistasis scan was limited to a small set of genes and it was feasible to run the full linear mixed model, therefore twin structure was modelled simultaneously with SNP effects to maximise power.

We have added the following sentence to the Methods: “This two stage procedure where relatedness was regressed out separately from v-eQTL mapping was adopted to make the full scan for v-eQTL computationally feasible.”

*What is the form of the interaction term? There are many ways to encode it which can involve more than one parameter for SNPs not in D'=1*.

We modelled epistasis as a multiplicative term in the dosages rather than a more general model, which would include factors such as recessive epistasis. This was for two reasons:

1) The interacting dosage model is consistent with expected expression under the assumption that cis interacting variants must share the same haplotype (recessive and dominant epistasis would require departures from what we expect is a reasonable model of a cis molecular interaction), and

2) Certain more general models of epistasis could manifest as an effect based on highly infrequent combinations of genotypes (such as both loci being minor allele homozygotes) which could produce significant findings based on very small numbers.

*Why use a non-parametric test for v-eQTL discovery and then a LMM for interaction? Although the data are quartile normalised, are the squared residuals and what is the effect of outliers*?

The squared residuals are not rank normalized: this is why a non-parametric test was applied as there are often departures from normality. An alternative would be to normalise the squared residuals and then apply linear regression, but we believe these two alternatives to be equivalent (as was argued in Battle et al. (2014), where a stepwise equivalent to the Spearman correlation test was required). When testing interactions, our approach is to follow the standard statistical methodology. Our solution to avoid false positives due to outlier effects is to use replication.

We also face the issue of heteroskedasticity, where the genotype dependent variance means that the axioms of linear regression do not hold. To ensure that our results are not caused by heteroskedasticity, we have considered various transformations to remove this issue and found the results to be robust. In particular, of the 131 statistically significant interactions in the GEUVADIS cohort, 126 are also significant when log transformed data is analysed (a typical way of accounting for heteroskedasticity). We now refer to this test in the Methods section.

*The conditional analysis presumably includes SNPs one-by-one to check the association holds – does imputation uncertainty matter here? Please also clarify why the influence of a second eQTL doesn't have an impact on the result*.

We assume the reviewers are discussing the analysis that investigated confounding by haplotype effects using the GEUVADIS dataset.

Although there is imputation uncertainty in the 1000 Genomes dataset, this is greatest for low frequency (below 1%) variants, whereas to explain away our observed epistatic interactions we would most likely require variants of higher allele frequency. Also, good haplotyping tagging is directly related to good imputation quality, thus we would expect such causative variants to have better imputation quality. However, we do recognise this as an issue and so have added the following caveat:

“The aim was for good characterisation of eQTL down to low frequency variants, though this is complicated by power and poorer imputation accuracy at such frequencies.”

With respect to the identification of eQTL, we have changed the manuscript. We now identify eQTL affecting expression in GEUVADIS by a forward stepwise scan with a threshold of 10^-5^ (this is more lenient than Bonferroni at the gene level, which varies from 3.1×10^-6^ to 10^-8^, and also the threshold applied in the GEUVADIS analysis, 6.6×10^-6^). Of the 131 genes, 103 had at least one eQTL, with numbers of eQTL ranging from 1 to 5. To discard haplotype effects as an explanation for the observed interaction we test each eQTL individually. If when controlling for *any* of the eQTL, the interaction is no longer significant, we discard this interaction. We believe this to be a conservative criterion for keeping interactions: in total 57 out of 131 survive this correction.

*In the main text: after identification of v-eQTL “to search for epistasis we scanned the cis windows for a second variant statistically interacting with each of the peak v-eQTL”. It would be helpful to include a mathematical description of the model*.

We have rewritten the Methods to give explicit mathematical formulae, which we agree gives greater clarity. In addition, we have made all code available so that the methodology can be implemented by anyone interested in doing so (in particular, for the GEUVADIS dataset for which data and methods are combined in an R workspace).

The epistasis section of the Methods has therefore been much enlarged, and a new Methods section “Equations” presents all linear mixed models used in this paper. Supplementary material has been uploaded where it is simple to repeat the replication analysis, and the TwinsUK scripts are provided so the methodology can be examined.